# Does Trade Policy Uncertainty Exacerbate Environmental Pollution?—Evidence from Chinese Cities

**DOI:** 10.3390/ijerph19042150

**Published:** 2022-02-14

**Authors:** Yiping Sun, Xiangyi Li, Tengyuan Zhang, Jiawei Fu

**Affiliations:** 1School of Business Administration, Hubei University of Economics, Wuhan 430205, China; ypsun82@hbue.edu.cn; 2School of Economics and Management, Hubei University of Technology, Wuhan 430068, China; 3School of Business Administration, Zhongnan University of Economics and Law, Wuhan 430073, China; tengyuanzhang5677@126.com (T.Z.); fjwzuel@163.com (J.F.)

**Keywords:** trade policy uncertainty, regional PM_2.5_ concentration, air pollution, China

## Abstract

Although the relationship between trade and environment has been widely discussed in past studies, trade policy has been in a state of continuous change in recent years. Previous studies have focused on the impact of trade opening or liberalization on the environment, ignoring discussion of the dynamic changes of trade policy. Therefore, it is very important to explore the connection between trade policy changes and environmental pollution for future environmental protection. In order to realize the in-depth study of this mechanism, the paper will try to solve the following three problems: (1) What is the relationship between change in trade policy uncertainty and China’s environmental pollution? (2) What is the mechanism by which trade uncertainty changes environmental pollution? (3) Due to China’s vast territory and regional differences, will changes in trade policy uncertainty have heterogeneous effects due to regional differences? To solve these problems, based on China’s accession to the WTO at the end of 2001, this paper, for the first time, uses PM_2.5_ concentration data of 246 prefecture-level cities in China to explore the impact of trade policy uncertainty on China’s environmental pollution, then we make an in-depth analysis of the impact path and heterogeneity of urban spatial distribution and city size. We found that, after China’s accession to the WTO, the growth rate of PM_2.5_ concentration reduced in cities with lower trade policy uncertainty and the inhibition effect was different due to the spatial distribution of city size. A further mechanism test shows that reduction in trade policy uncertainty can improve environmental pollution through industrial, structural and technological effects.

## 1. Introduction

Environment is the basis for human survival and development and air pollution is an important factor in environmental problems. In recent years, it has become a problem that all countries must pay attention to and needs to be solved. Air pollution caused nearly half a million deaths of infants around the world in 2019, most of them in developing countries, according to the “2020 State of The World’s Air Report”, which notes that the global air quality situation remains severe. In 106 monitored countries, only 24 met the World Health Organization’s (WHO) annual guidelines for PM_2.5_ in 2020. In addition, the United Nations Environment Programme (UNEP) points out that emissions of greenhouse gases are far higher than the proposed standards of the “United Nations framework convention” on climate change, and cause global warming, ozone depletion, atmospheric acidification and a series of difficult problems.

In the early days of their economic development, many countries tend to pay more attention to economic indicators and ignore environmental protection, leading to serious environmental problems which were more common in developing countries, and China has such problems. In 2001, China joined the World Trade Organization (WTO) to enjoy the permanent most-favored-nation (*MFN*) optimal tariff treatment, and China’s foreign trade entered a new track of rapid development. From 2002 to 2012, the average annual growth rate of foreign trade was more than 20%, and the total volume of trade increased from 5.14 trillion Yuan to 24.42 trillion Yuan, making China the world’s largest trading country for the first time (Sources of data: State Statistics Bureau). From 2013 to 2018, China’s rapid development made a huge contribution to world economic growth, with a growth rate of 28.1%, ranking first in the world (Sources of data: http://finance.people.com.cn/n1/2019/1012/c1004-31395472, accessed on 30 October 2019). The development of trade has brought sufficient impetus to China’s economic recovery and prosperity, but China has also borne a huge environmental price during this process. For a long time, local government’s performance evaluation system was not sound, with production competition and GDP as the core goal, and this extensive economic development has caused great damage to the ecological environment. There has been rapid economic growth at the cost of environment. Such behavior has brought huge “sequelae”, causing irreversible negative effects on people’s life which will endure for a long time into the future. At the beginning of 2013, China suffered the worst haze weather in history, affecting 25 provinces and more than 100 large and medium-sized cities. The average number of haze days nationwide was as high as 29.9 days, and in some cities in northeast China experienced PM_2.5_ concentration reached an astonishing 1000 ug/m^3^, seriously affecting people’s daily life and economic development (Sources of information: Chinese Industry Research Network: http://www.cnr.cn/ accessed on 30 December 2013). In June of that year, the Chinese government issued “The Action Plan for The Prevention and Control of Air Pollution”, determined to thoroughly control air pollution. In 2017, at the 19th National Congress of the Communist Party of China (CPC), an important working conference held every five years, the Chinese government decided to make ecological progress one of the three critical battles in order to achieve a moderately prosperous society in all respects. China formally enacted and implemented “the Air Pollution Prevention and Control Law of the People’s Republic of China” in 2018. The 75th Session of the United Nations General Assembly was held on 22 September 2020, and various countries proposed environmental protection proposals. The Chinese side proposed as follows: “China will increase its nationally determined contribution, adopt more effective policies and measures, strive to peak carbon dioxide emissions by 2030, and strive to achieve carbon neutrality by 2060”, calling on people around the world to build an ecological civilization. 

At the same time, the world economy slowed down, and the developed economies led by the United States, Japan and the European Union, as well as emerging economies represented by India, Brazil and Russia, all faced economic growth bottlenecks. Since 2016, global trade protectionism, hegemonism and nationalism have flourished, and the international economic order is in flux. At the beginning of 2020, the global economy was hit hard by the outbreak of COVID-19. Some countries continued to disrupt the international economic order through sanctions and power play, pushing trade uncertainties to a new high level. The impact of the epidemic on the global economy is lasting and profound. In 2021, some developed countries have participated in incredible behaviors, such as nuclear sewage discharge. The re-emergence of economic considerations at the expense of the environment in countries with more advanced and sound environmental awareness and standards has given us pause for thought. In the post-COVID-19 era, the international economic and trade environment has undergone major changes, and trade policy uncertainties are rising. It is an unsolved puzzle as to whether the environment, as the “cheapest” resource, will be “put into production” again. Already, partly at the repeated expense of the environment internationally, in such a grim international economic and trade situation for the environment, China has an exemplary role in how to stick to the operational concept of “harmony between man and nature”, correctly handle its environmental pollution problem for the benefit of world environmental pollution prevention and global economic recovery. Therefore, it is necessary to clarify the internal mechanism between change in trade policy uncertainty and environmental pollution. The completed research is divided into the following three problems. Based on the natural experiment of China’s accession to the WTO, this paper makes an in-depth analysis of the change in trade policy uncertainty and environmental pollution from the following three aspects. (1) What is the relationship between change in trade policy uncertainty and China’s environmental pollution? (2) What is the mechanism by which trade uncertainty changes environmental pollution? (3) Due to China’s vast territory and regional differences, will changes in trade policy uncertainty have heterogeneous effects due to regional differences? The resolution of these three issues will give us a systematic understanding of the relationship between trade policy uncertainty and environmental pollution. At the same time, it can also open up a new perspective for this kind of research. At the present stage, most studies on trade policy uncertainty focus on enterprise behavior, and no author has conducted systematic research on its relationship with environmental problems, but this is indeed a very important direction, and in-depth research will be conducted and a demonstration role offered.

The rest of this paper will be arranged as follows. The second part is the literature review on trade policy uncertainty and environmental pollution; the third part is the analysis framework, including problem analysis, data, estimation strategies and variables; the fourth part contains the empirical results and analysis. This part first analyzes the basic regression results and carries out a series of validity and robustness tests to ensure that the conclusions are not disturbed. The fifth part is the heterogeneity analysis, which studies the impact of trade policy uncertainty on pollution in different cities. The sixth part is the analysis of the mechanism linking trade policy uncertainty and environmental pollution; the seventh part is the conclusion.

## 2. Literature Review

Concerning studies on environment and trade, many scholars have made great contributions to trade openness and trade liberalization, but few scholars have conducted systematic studies on the subject of trade policy uncertainty and environmental pollution, so this paper intends to conduct in-depth studies on this issue. The literature closely related to this paper can mainly be divided into the following two branches.

The first branch is the study of trade policy uncertainty. Existing studies on this impact mainly focus on enterprise behavior. First is import and export behavior. Based on the study of Australia’s multilateral policy commitment, Handley [1] found that when the uncertainty of trade policy decreases, enterprises’ exports will be positively affected. Handley and Limao [2] studied the impact of the reduction of trade policy uncertainty on enterprises’ exports after Portugal joined the European Community, and found that the reduction of trade policy uncertainty had a positive impact on enterprises’ export, and more new enterprises participated in the export market. The studies of Osnago et al. [3] and Kim [4] show that increased uncertainty in trade policy hinders enterprises’ exports. Handley and Limao [5] discussed the reduction in trade policy uncertainty and enterprises’ export issues based on China’s accession to the WTO in the same way and reached the same conclusion. In addition, Feng et al. [6] conducted a study on changes in trade policy uncertainty and export decision-making of Chinese enterprises, and found that product quality and price are the channels through which the reduction of trade policy uncertainty improves the reallocation efficiency of export activities among enterprises. Second is the study of enterprises’ innovation behavior. Akcigit et al. [7] and Liu and Ma [8] study the relationship between the uncertainty of trade policy and enterprise’s innovation. They found that there was a negative relationship between trade policy uncertainty and enterprise’s innovation. That is, the rise of trade policy uncertainty has a negative effect on enterprises’ innovation. Third is the study of enterprises’ investment behavior. Shepotylo and Stuckatz [9] found that high trade policy uncertainty had a negative impact on enterprises’ imports, exports and investment. Pierce and Schoot [10] also studied the relationship between trade policy uncertainty and enterprises’ investment and found that reduction of trade policy uncertainty significantly reduced the level of investment by US enterprises. Later, Andrew et al. [11] found in their study that trade policy uncertainty reduced enterprise investment, mainly because the export investment of new enterprises or new products was irreversible. Caldara et al. [12] investigated the impact of trade policy uncertainty on enterprises’ capital expenditure and found that increase in trade policy uncertainty in the United States inhibited increase in enterprises’ physical investment and had a more obvious impact on export enterprises. 

In addition to enterprise behavior, some scholars have explored other aspects. For example, Pierce and Schoot [13] found that the reduction in trade policy uncertainty caused by the granting of PNTR by the United States to China led to a decline in the number of manufacturing jobs in the United States. Facchini et al. [14] studied the impact of the reduction in trade policy uncertainty on the flow path of China’s population. Li et al. [15] found that the Chinese government tends to increase subsides to energy firms when trade policy uncertainty rises, and political connections play an active role in reinforcing the causal effect.

The second approach is the impact of trade on the environment. At present, there are three viewpoints. First, trade opening can improve environmental quality. Grossman and Krueger [16] divided the impact of international trade on the environment into three aspects: scale effect, industrial structure effect and technology effect. Based on this framework, Antweiler et al. [17] found that trade has a positive effect on environmental improvement in general. Frankel and Rose [18] studied the relationship between trade liberalization and the environment and found that trade liberalization improved the environment. Next, Mcausland [19] and Beladi and Oladi [20] found that the technological effect of trade liberalization strengthened research and development into cleaner production technology. This is helpful for the diffusion and spillover of emission reduction technology, thus reducing environmental pollution. Cui et al. [21] conducted a specific study on China’s environmental problems and found that China is a net exporter of implied energy emissions, and raising export tariffs on energy-intensive industries in China will help improve environmental quality.

Second, trade openness has a negative impact on environmental quality. At the beginning of the 20th century, Chichilnisky [22] began to discuss the relationship between trade and environment, and found that trade opening enabled developing countries to export environment-intensive products at low prices, which adversely affected their own environmental quality. In the 21st century, the research on this problem can be roughly divided into two categories. First, concerning research on trade and environmental issues in specific countries, Stahls et al. [23] studied Finland’s environment and trade, with CO_2_ emissions as an environmental measurement index. They found that export was the main source of environmental deterioration. Poncet et al. [24] used per capita SO_2_ emissions as pollution index to study China’s environmental problems and found that different types of enterprise had different but negative effects on the environment, and foreign-funded enterprises had more significant negative effects than domestic enterprises. Second are studies on trade and environmental issues in specific types of country, including developing and developed countries. Lin [25], Aklin [26] and Andersson [27] found that international trade had negative impacts on air quality in most developed and developing countries. Regarding high-income and low-income countries, Le et al. [28] used PM_10_ emission data and found that, after trade liberalization, environmental pollution in high-income countries was alleviated, but pollution levels in middle and low-income countries were higher. Overall, pollution levels at the global level increased.

Third, the impact of trade openness on the environment cannot be clearly differentiated. Stevens [29] divided the effects of trade liberalization on the environment into five mechanisms: trade scale, resource allocation efficiency, environmental regulation, output structure and production technology. These mechanisms interact with each other, and the specific influence direction needs specific analysis. Matthew and Robert [30] proposed the concept of scale technology effect, which in essence is the combination of scale effect and technology effect. They found that trade liberalization would result in the discharge of some pollutants, but at the same time new pollutants would also be generated, so it was impossible to judge the specific trend of environmental pollution. Baek [31] used SO_2_ emission data of 50 countries for analysis and found that high trade openness has different effects on different types of country; specifically, air quality in developing countries is negatively affected while in developed countries it is positively affected. Managi et al. [32] argue that the impact of trade opening on the environment varies with the choice of pollutant measures and by country. The study found that trade reduced SO_2_ and CO_2_ in OECD countries, but conversely in non-OECD countries.

The contribution of existing studies on trade opening, trade liberalization and environmental issues has provided a great deal of the theoretical basis and inspiration for this study. However, previous studies on trade policy uncertainty are insufficient and the novel contribution of this paper is mainly reflected in the following aspects: (1) The research perspective is expanded. At present, most studies on trade policy uncertainty focus on enterprise behavior. Instead of continuing to study enterprise behavior, trade policy uncertainty is combined with environmental issues, which opens up a new horizon for research in this field. In addition, previous literature on environment and trade issues mainly focused on the perspective of trade openness and trade liberalization, but this paper focuses on change in trade policy uncertainty, which provides a new perspective for the analysis of environmental issues. (2) The accuracy of research data is improved. In the past, literature in the field of trade and environment mainly used data at the national level, while only a few studies used data at the provincial and regional levels, and data at the regional level were relatively limited. In addition, carbon emission and other indicators are mostly used to measure environmental pollution [33,34,35], and only a small number of more accurate indices of emerging pollutant PM_2.5_ are used to study environmental pollution. In this paper, PM_2.5_ data of 246 prefecture-level cities in China from 1998 to 2007 are used to conduct in-depth research on this issue. This is the first time that such a long time span and wide coverage data are used regarding this issue. (3) In-depth analysis of the content. Previous literature studies mainly focus on global and regional organizations, developing countries and developed countries, and there are only a few, and not very detailed, studies on specific countries. This paper takes China as the research object. We not only analyze the relationship between trade policy uncertainty and environmental pollution, but also analyze the pathway and the heterogeneity of the impact effect from spatial distribution characteristics and city size.

## 3. Analytical Framework

### 3.1. Analysis

One of the major changes in uncertainty regarding China’s trade policy that can be studied is China’s accession to the WTO at the end of 2001. Before joining the WTO, China was only granted temporary Normal Trade Relations (NTR) status by the US, which was subject to review by the US Congress every year. If it failed to pass the review, it would face high tariffs. In history, the US Congress has passed three bills to revoke China’s NTR, which has brought great uncertainty to China’s trade environment. After China’s accession to the WTO granted permanent normal trade relations (PNTR) status, there is no longer a need to accept the review of the U.S. congress each year, and this reduction in trade policy uncertainty creates a stable trade environment for Chinese companies and foreign trade and business continues to grow steadily. This article carries out research based on China’s accession to the WTO. In order to clarify the detailed situation of local pollution after this accession, the author plotted the trend chart of PM_2.5_ concentration in each province of China from 1998 to 2007. 

Figure 1 shows the change of PM_2.5_ concentration in each province after China’s accession to the WT. We can see the trend of PM_2.5_ concentration in each province in China from 1998 to 2007. On the whole, PM_2.5_ concentration in all provinces showed an upward trend. In particular, after China’s accession to the WTO in 2001, the rate of increase in northeast China and central and western China was high, and some provinces with high PM_2.5_ concentration even doubled their output. It can be seen that China’s environmental situation has become increasingly severe after China’s accession to the WTO and a large number of provinces are falling into the environmental pollution dilemma. However, Figure 1 can only help us understand the general situation of China’s environment, and it cannot clearly judge the relationship between changes in trade policy uncertainty and environmental pollution. Therefore, cities in China are divided into two groups in this paper. The control group consists of cities with low trade policy uncertainty, and the treatment group consists of cities with high trade policy uncertainty. According to the divided groups, PM_2.5_ concentration changes are explored. Since China joined WTO at the end of 2001, 2002 and subsequent years are regarded as the display period for the event effect, as shown in Figure 2.

According to Figure 2, it can be found that, before 2002, PM_2.5_ concentration in the two groups of cities had the same upward change trend on the whole. After 2002, PM_2.5_ concentration in the two groups of cities showed an upward trend on the whole, but there was a significant difference in the rate of increase between the two groups of cities, and the gap between the two groups gradually widened. The rising rate of the control group was faster, while the rising speed of urban pollution level in the treatment group decreased significantly comparatively. It can be seen that cities with large changes in trade policy uncertainty have a slower increase rate of PM_2.5_ concentration. Before the policy implementation year, the pollution level of the two groups of cities showed a relatively consistent change trend. After the policy implementation year, there is a significant difference in the change in the pollution level of the two groups of cities, which is preliminarily determined to meet the assumption of a parallel trend. In the later empirical research, dynamic effects will be used to verify this trend more rigorously. We speculate that the reduction in trade policy uncertainty will have a positive effect on the environment. Based on such speculation, this paper decides to use the following model for research.

### 3.2. Model and Variable

In order to identify the impact effect of trade policy uncertainty accurately and effectively on the environment, this paper adopts the difference-in-difference method (DID for short) of empirical research, and the specific model is set out as follows:(1)$${\mathrm{PM}}_{2.5\;jt}=\alpha +\beta {\mathrm{TPU}}_{jt}*Post02_{t}+\gamma X_{jt}+\delta_{jt}+\varepsilon_{jt}\;$$
where ${\mathrm{PM}}_{2.5\;jt}$ represents the PM_2.5_ concentration of city j in t years, which is used to measure the urban haze pollution level. ${\mathrm{TPU}}_{jt}$ represents the trade policy uncertainty of city j in t year; $Post02_{t}$ represents the dummy variable of the year when China joined the WTO. The year is greater than or equal to 2002 when $Post02_{t}\;$= 1, otherwise $Post02_{t}$ = 0. ${\mathrm{TPU}}_{jt}*Post02_{t}$ is the core explanatory variable of the econometric regression model, and its coefficient β measures the impact of trade policy uncertainty on quality of environmental development. $X_{jt}$ represents the control variable, $\delta_{jt}$ represents the fixed effect of city and year, and $\varepsilon_{it}$ represents the error term. At the same time, the standard error is clustered at the city level by regression. 

The explained variable and the measurement of explanatory variables are as follows:

1.PM_2.5_ concentration. The measurement index of environmental pollution is different from that of the conventional air pollutants such as SO_2_, CO_2_, CO, API and PM_10_ adopted in most literature. In this paper, PM_2.5_ concentration, which is a concern for all sectors of society, is selected for measurement purposes. The PM_2.5_ concentration data used in this paper covers 246 cities, covering most cities in China, and the time span is as long as 10 years, which can provide a solid data basis for accurate screening of the impact of trade policy uncertainty on environmental pollution.2.Trade policy uncertainty (${\mathrm{TPU}}_{jt}$). At present, there are three main methods to measure the uncertainty of trade policy: first is the text extraction method. Baker et al. [36] measure the uncertainty of trade policy by using the frequency of relevant words (“Congress”, “deficit”, etc.) Pierce and Schott [13] use the difference between normal trade partnership tariffs and abnormal trade partnership to measure the impact of the permanent normalized trade partnership granted by the United States to China. Handley and Limao [5] used the change degree of tariffs (the ratio of tariffs in abnormal trade partnerships to those of tariffs in normal trade partnerships) to measure trade policy uncertainty. The first method requires high labor costs and inaccurate estimation results, while the difference between the second and the third method is small. Therefore, this paper chooses to refer to Pierce and Schott’s [13] method to calculate the trade policy uncertainty index. The specific measurement method is as follows:(2)$${\mathrm{TPU}}_{i}=tariff_{i}^{COL2}-tariff_{i}^{MFN}$$

We construct the non-tariff rate gap using non-tariff rate data from 1999, prior to China’s accession to the WTO. Among these, $tariff_{j}^{COL2}$ represents the tariff for the abnormal trade relationship of product *i*, ${\mathrm{and}\;tariff}_{j}^{MFN}$ represents the most-favored national tariff for product *i*. 

Next, the author refers to the method of Bartik [37] and uses the Chinese Customs trade database, which contains the quantity of exported goods from enterprises to the United States. This paper uses this data mainly to calculate weight. In particular, the product shares in China’s export basket observed from 1997 to 1999 (before China’s accession to the WTO) are used as weights to measure trade policy uncertainty at the regional level. Specific methods are as follows:(3)$${\mathrm{TPU}}_{j}=\underset{i}{\sum }\frac{Exp_{ij}}{Exp_{j}}*{\mathrm{TPU}}_{i}$$
where $Exp_{ij}$ is the product *i* exported from city *j*, and $Exp_{j}$ is all the products exported to city *j*.

3.Control variables ($X_{jt}$). Combined with existing studies, this paper also controls a set of urban characteristic variables in the benchmark regression model to mitigate omission bias as much as possible. This group of variables include capital–labor ratio (K/L), population density (Pop), proportion of secondary production (Manu), foreign direct investment (FDI), per capita GDP (P. C. GDP), land area (Land), and industrial output (Output). The capital–labor ratio is expressed by the fixed capital input and the proportion of employees. Population density refers to the number of people per unit area; the proportion of secondary industry is expressed by this proportion in GDP; foreign direct investment is measured as such; gross industrial output value shall be measured by this value above the designated level.

The data and statistical description of the variables explained above, core explanatory variables and a series of control variables are shown in Table 1.

### 3.3. Data Description

The data used in this paper are divided mainly into three sets. The first set is the 1989–2001 import tariff data for the United States collated by Romalis. This database records the HS8-bit code tariff rates of the United States for normal trading partner countries (*MFN*) and non-normal trading partner countries (COL2) in detail. From this data, the TPU at city level in 1999 could be calculated (Romalis collates tariff data from his website: http//www.johnromalis.com/publications, accessed on 30 December 2002). The second set are regional PM_2.5_ concentration data, extracted by The Socioeconomic Data and Applications Center (SEDAC) of Columbia University using NASA satellite data. This includes PM_2.5_ concentration data from 343 prefecture-level cities in China from 1998 to 2016 and 6-digit city codes were combined, with the first set based on the first 4-digit city codes. This paper only uses the data from 1998 to 2007 for the following reasons: first, during this time period, China entered the WTO and experienced a sharp reduction of trade policy uncertainty, which provides a good quasi-natural experiment from which to study trade policy uncertainty and environmental pollution. Second, the research method used in this paper is DID. In addition to meeting the parallel trend assumption before policy impact, the use of this method also has an important premise in identifying the causal effect of the difference-in-difference strategy. This premise requires that after the implementation of the evaluation policy, no other policy has a systematic heterogeneity effect on the outcome variables of the treatment group and the control group, thus requiring that the span of the study sample period after the implementation of the policy should not be too long. The date of the existing data is up to 2016. 2013 is the first year of pollution prevention and control in China. Since 2013, the Chinese government has begun to realize the severity of the environmental pollution problem and has made great efforts to control environmental pollution. Therefore, data after 2013 do not meet the requirements. In addition, the sample time span before China’s accession to the WTO was only 3 years. If the sample deadline is 2013, the sample time span after China’s accession to the WTO is 13 years, which may affect the accuracy of causal identification. In addition, the sample range of international top scholars such as Brandt et al. [39] regarding this kind of problem is limited to 2007, so the sample range of this paper is limited to 1998–2007. However, in order to prove the reliability of the conclusion, the author extended the time interval to 2013 in the robustness test. The third set is the Statistical Yearbook of Chinese Cities, which includes some control variables at the city level: foreign direct investment, population density, land area, proportion of secondary industry, capital–labor ratio, industrial gross output value and per capita GDP. The first two sets of data are integrated with six-digit city codes and years.

## 4. Empirical Results and Analysis

### 4.1. Baseline Regression

The benchmark regression results based on Equation (1) are shown in Table 2. In Column (1), only core explanatory variables ${\mathrm{TPU}}_{jt}*Post02_{t}$ are included, except for the city fixed effect and year fixed effect. The results show that the coefficient of core explanatory variables is significantly negative at the level of 5%, indicating that after China’s entry into the WTO, cities with a greater reduction in trade policy uncertainty in the treatment group (i.e., cities with previously high trade policy uncertainty) had a lower increase rate in PM_2.5_ concentration than cities with a smaller reduction in trade policy uncertainty in the control group (cities that had previously had low trade policy uncertainty), and reduced trade policy uncertainty can obviously reduce environmental pollution. Columns (2)–(4) gradually add city-level control variables, including foreign direct investment, population density and proportion of secondary industry. The results show that these three variables promoted the increase of PM_2.5_ concentration, but the effect was not significant. Then, the city-level control variables such as capital–labor ratio, per capita GDP, land area and total industrial output value are successively added in columns (5)–(8). It can be found that capital labor ratio is significantly positive at the 5% level for the concentration of PM_2.5_ effect, which shows that the capital-labor ratio has a significant positive effect on environmental pollution. It is also found that per capita GDP has a significant negative effect on environmental pollution at the significant level of 1%. In addition, land area and industrial output value have no significant impact on PM_2.5_ concentration. After the addition of control variables, the influence of core explanatory variables${\;\mathrm{TPU}}_{jt}*Post02_{t}$ on the explained variables did not change significantly, which further proves the reliability of the conclusion that the reduction of trade policy uncertainty can significantly reduce air pollution. 

### 4.2. DID Validity Test

Although DID can solve the endogenous problems that may exist in policy evaluation, some important assumptions need to be satisfied. This part will verify the reliability of the results obtained through validity and robustness tests such as dynamic effect, expected effect and placebo test.

#### 4.2.1. Dynamic Effect

In order to further study the validity of the conclusion regarding the impact of trade policy uncertainty on air pollution, it is necessary to study the dynamic effect of trade policy uncertainty on air pollution. In this paper, the core explanatory variable ${\mathrm{TPU}}_{jt}*Post02_{t}$ is replaced by the multiplication of ${\mathrm{TPU}}_{jt}$ and the dummy variables of each year from 1998 to 2007 (1998 as the base year), and is added into the regression equation to obtain the DID extended model as shown in formula (4). Column (1) of Table 3 reports the regression coefficients of interaction terms between TPU and dummy variables in each year. The results show that the regression coefficients of interaction terms are not significant before China’s accession to the WTO, which indicates that before this date the treatment group and the control group accord with the parallel trend assumption, and the regression coefficients of interaction terms become significant from 2002. This shows that the reduction of trade policy uncertainty after China’s accession to the WTO has a significant effect on environmental pollution.
(4)$${\mathrm{PM}}_{2.5\;jt}=\alpha +\sum_{r=1999}^{2007}\beta_{r}{\mathrm{TPU}}_{jt}*year_{t}+\gamma X_{jt}+\delta_{jt}+\varepsilon_{jt}\;$$

#### 4.2.2. Expectation Effect

China had been negotiating for 15 years before formally joining the WTO. This may have enabled domestic enterprises to prepare for China’s accession to the WTO in advance and adjust their production and operation behavior according to expectations. Therefore, this paper further added the interaction term of ${\mathrm{TPU}}_{jt}$ and the Prereform of the year before China formally joined the WTO into the regression Equation (5).
(5)$${\mathrm{PM}}_{2.5\;jt}=\alpha +\beta {\mathrm{TPU}}_{jt}*Post02_{t}+{\mathrm{TPU}}_{jt}*Prereform+\gamma X_{jt}+\delta_{jt}+\varepsilon_{jt}\;$$

The corresponding regression results are reported in column (2) of Table 3. It can be seen that the regression coefficient of trade policy uncertainty is basically unchanged after the addition of the expectation term, and the coefficient of expectation term is not significant, which indicates that China’s WTO accession policy is exogenous, ensuring the accuracy of the estimation results. 

#### 4.2.3. Placebo Test

In the quasi-natural experiment of this paper, on the one hand there is a high correlation between the development level of each city and its trade, and the government will also display certain tendencies in the process of formulating trade policies. On the other hand, although we have controlled feature variables at the city level in Formula (1), it is not enough to control all features. Limited by data, some features cannot be observed, so the results of this paper may be interfered with by these non-observed factors. In this regard, we adopt the following ideas to indirectly test whether these omitted non-observation factors will have an impact on the estimated results. Firstly, we obtain the estimated coefficient $\hat{\beta }$ expression of the ${\mathrm{TPU}}_{it}*Post02_{t\;}$ based on Equation (1): (6)$$\hat{\beta }=\beta +\lambda \cdot \frac{c\mathrm{ov}({\mathrm{TPU}}_{i01}*Post02_{t}|w)}{var({\mathrm{TPU}}_{i01}*Post02_{t}|w)}$$
where w represents all control variables, and if λ=0, non-observational factors will not affect the estimated results and $\hat{\beta }$ bearing is unbiased, but this does not directly verify that λ is 0. As an indirect consideration, λ=0 can be proved if a variable can be found in place of ${\mathrm{TPU}}_{it}*Post02_{t}$ and if this variable theoretically has no real impact on the corresponding ${\mathrm{PM}}_{2.5\;jt}$ (meaning β=0), and the assumption of $\hat{\beta }$ bearing zero can be assumed. Therefore, according to the idea of Liu and Lu [40], we performed a placebo test to makes sure the policy impact variable of China’s accession to the WTO is random (500 times in the random sampling process), such random processing can ensure no effect on ${\mathrm{PM}}_{2.5\;jt}$, that is, $\beta_{\mathrm{random}}$ = 0. In this case, we can also estimate the mean of ${\hat{\beta }}_{\mathrm{random}}$ and show the distribution of the estimated 500 ${\hat{\beta }}_{\mathrm{random}}\;$in Figure 1. As can be seen, the estimated coefficient ${\hat{\beta }}_{\mathrm{random}}$ obtained after random processing is distributed around 0, which is significantly different from the benchmark regression coefficient. Therefore, λ = 0 can be backward deduced. Therefore, the observed characteristics have little impact on the estimated results, and the previous estimation is valid. The placebo test results are shown in Figure 3.

### 4.3. Robustness Test

First, this paper uses the method of Handley and Limao [5] to re-calculate the ${\mathrm{TPU}}_{j}\;$in 1999. The specific measurement method is as follows:(7)$${\mathrm{TPU}}_{i}=1-{\left(tariff_{i}^{COL2}/tariff_{i}^{MFN}\right)}^{-\delta }\;$$

This paper takes δ = 2, δ = 3 and δ = 4, respectively, to further verify the stability of the results of this trade policy uncertainty measurement method (TPU measurement method at the city level remains unchanged). Columns (1)–(3) of Table 4 report the estimation results of regression. Second, on the basis of controlling for city fixed effect and year fixed effect, and adding province fixed effect, column (4) of Table 4 reports the estimation results of regression. Third, the clustering standard error is adjusted from the city level to the province level. Column (5) of Table 4 reports the estimation results of regression. Thirdly, the samples were tail-tailed with 5% of upper and lower values, with truncated data with 5% of upper and lower values, and the samples were expanded to 2013 for testing. Columns (6)–(8) of Table 4 report the estimation results of regression. Fourth, referring to Facchini et al. [14], the author uses an unweighted (simple average) measure of TPU and constructs an alternative weighted TPU, which uses the data of China’s exports to the world instead of just exports to the US to calculate the weight. Columns (9)–(10) of Table 4 report the estimated results of regression. In column (11), total energy consumption (coal) is added to improve the reliability of regression results (energy consumption from the China energy statistical yearbook). Obviously, the coefficient and significance of the core explanatory variable have always maintained good consistency, indicating that, after a series of robustness tests, the coefficient of the core explanatory variable (cross term) is still significantly negative, indicating that the conclusion, i.e., that the reduction of trade policy uncertainty will improve environmental quality, is stable and reliable. 

### 4.4. Bartik IV

In order to better avoid the endogenous problem in measurement identification, a bartik instrument (the product of the first-order lag trade policy uncertainty TPU_(*j*, *t* − 1), and the first-order difference ΔTPU(*t*, *t* − 1) in time) is constructed based on the practice of Bartik [37], and then the 2SLS two-stage instrumental variable estimation is carried out. In the first stage regression results in Table 5, the Bartik instrumental variable is significantly positive to the ${\mathrm{TPU}}_{jt}*Post02_{t}$ regression coefficient, which shows that the instrumental variable is effective. In addition, in the two-stage regression, the statistical value of F is 47.09, which is greater than the critical value of stock-Yoo weak instrumental variable 10, i.e., it has passed the weak instrumental variable test. It can be seen from the regression results of Bartik instrumental variables in this part that the results remain robust and reliable after considering the endogenous problem.

## 5. Heterogeneity Impact Analysis

In this part, the heterogeneity of the model’s results is analyzed from two aspects: the spatial distribution characteristics of environmental pollution caused by trade policy uncertainty and the difference in city size. 

### 5.1. Spatial Distribution Characteristics

Referring to the practice of Fan et al., Fan et al., Li et al. and Wang [41,42,43,44], this part divides the total sample into three regional sub-samples, namely, the eastern sample, the central sample and the western sample (Eastern provinces: Hebei, Liaoning, Jiangsu, Zhejiang, Fujian, Shandong, Guangdong and Hainan; provinces of central cities: Jilin, Anhui, Shanxi, Jiangxi, Henan, Hubei, Hunan, Heilongjiang; provinces (autonomous regions) of western Cities: Yunnan Province, Inner Mongolia Autonomous Region, Sichuan Province, Ningxia Hui Autonomous Region, Guangxi Zhuang Autonomous Region, Xinjiang Uygur Autonomous Region, Gansu Province, Guizhou Province, Shaanxi Province and Qinghai Province.). Because the foreign trade system of Hong Kong, Taiwan and Macao are different, they are not taken as research objects in this study. The research method in this part is consistent with the previous study of the heterogeneous impact of trade policy uncertainty on urban environmental pollution in different geographical locations. Columns (1)–(3) of Table 6 report the estimation results of regression, which showed that the core explanatory variables${\;\mathrm{TPU}}_{jt}*Post02_{t}$ in western China were significantly negative at the level of 10%, while the core explanatory variables${\;\mathrm{TPU}}_{jt}*Post02_{t}$ in eastern and central China were not significant. The possible reason is that the central and eastern regions of China include the Yangtze River Delta, Pearl River Delta, Bohai Economic Circle and other frontier demonstration areas of economic development. As the earliest area of China to realize reform and opening up, the degree of opening up trade has been very deep. Earlier established foreign trade channels and external communication, attracting a large amount of external investment, has brought advanced production experience and technology, and due to the collaborative efficiency of innovative cities in the Middle East [45], has promoted the development of environmental protection technology and industry, forming a relatively successful development of environmental awareness, able to quickly adhere to international standards. Therefore, after China’s accession to the WTO and the corresponding drop in trade policy uncertainty drop, there is no significant effect on the environment. In contrast, the western region, located in the interior, does not have good foreign trade conditions, has some high emissions and highly polluting industry to survive in competition with the eastern region. The western region become the scene of temporary resettlement, which coupled with abundant natural resources in western China caused the local government, in order to further economic development, to ignore the environmental protection problem in the process of mining resources. After China’s accession to the WTO, the infrastructure and foreign trade channels in the western region have been constantly improved, and the government has received corresponding policy support, prompting the reform and transformation of economic development in this region [46]. Rapid strides in foreign trade and the introduction of foreign investment has promoted the adjustment of industrial structure in western China, and highly polluting industry has gradually reduced in proportion of gross value of industrial output, information, high technology, production-oriented service industrial development, local government development, idea transformation, causing environmental protection consciousness to gradually strengthen, therefore effectively reducing atmospheric pollution. If the opening-up of the western region is further strengthened, it will not only promote the economic development of the region, but also reduce environmental pollution in the region to a certain extent. 

### 5.2. City Size

As an important aspect of urban difference, urban size may have an important impact on environmental pollution. This paper classifies first-tier cities, second-tier cities and third-tier cities as medium and large cities, and other cities as small cities according to the Notice of The State Council on Adjusting the Standards for Classifying city sizes. Columns (4)–(5) of Table 6 show the hierarchical distribution characteristics of environmental pollution caused by the reduction in trade policy uncertainty. The coefficient of core explanatory variable ${\mathrm{TPU}}_{jt}*Post02_{t}$ of small cities is significantly negative at 1% level, while that of large and medium-sized cities is not significant, indicating that the reduction in trade policy uncertainty significantly reduces the environmental pollution of small cities, but has no significant effect on the reduction in environmental pollution of large and medium-sized cities. This may be related to different scales of the development of city consciousness. Usually, large and medium-sized cities in terms of funding, policy, talents, information and innovation have a huge advantage [47], the development of city planning is more advanced, and at the same time, the standards of the population are higher, with higher requirements for environmental quality. In addition, big cities, relative to small cities, maintain closer contact with the outside world, earlier contact with internationally leading environmental protection standards, and relatively strong environmental awareness, so some of the more polluting industries find it difficult to survive in large-scale cities, and generally adopt the development strategy of moving to small cities. However, most small cities take economic growth as their first development goal, and their environmental awareness is relatively weak, which provides a certain space for the transfer and development of polluting industries. With increasing public awareness of environmental protection, this space for polluting industries in small cities is squeezed, and the reduction of trade policy uncertainty has a more significant effect on these small cities. 

## 6. Mechanism Test

An important finding of this study is that the reduction in trade policy uncertainty significantly reduces environmental pollution. So how does this come about? In order to explore this problem, based on the above theoretical analysis, this paper verifies the industrial structure effect and technological effect, respectively. The industrial structure effect is measured by the proportion of tertiary industry in GDP, and the technology effect is measured by the financial expenditure on scientific undertakings. Th test of influence is carried out by referring to the mechanism test method of Liu and Ma [8]. 

### 6.1. Effect of Industrial Structure

Column (1) of Table 7 reports the estimation results, with the effect of industrial structure as the dependent variable. As can be seen from column (1) of Table 7, the coefficient of the core explanatory variable ${\mathrm{TPU}}_{jt}*Post02_{t}\;$is significantly positive at the level of 10%, indicating that trade policy uncertainty can reduce environmental pollution by significantly promoting the transformation of the industrial structure of the city in the direction of low pollution. A possible reason is that, when China joined the WTO, on the one hand China and the world became closer, environmental protection consciousness gradually improved, and due to relevant environmental standards in line with international standards, the development of industrial energy development was restricted, along with the elimination of pollution intensive enterprises. At the same time as trade was continuously expanding, the influence of the income effect appeared, and consumers increased their demands for quality goods, paying more attention to the environmental friendliness of consumer goods, i.e., the green degree of products, which would have an impact on the decision-making of relevant enterprises [48]. Such changes would bring internal optimization and external development to secondary industry secondary industry is gradually transforming into service industry), thus reducing environmental pollution. On the other hand, with the opening up of trade, service trade increased rapidly, stimulating the development of China’s service industry. In addition, China’s economic development level increased year by year, changing the economic growth mode, with primary and second industry gradually transforming to services, with service industry development entering a “golden period”, thus reducing environmental pollution. 

### 6.2. Technique Effect

Column (2) of Table 7 reports the estimation results, with technological effect as the dependent variable. As can be seen from column (2) of Table 7, the coefficient of the core explanatory variable ${\mathrm{TPU}}_{jt}*Post02_{t}\;$is significantly positive at the level of 5%, which indicates that trade policy uncertainty can significantly promote the improvement of technological level to reduce haze pollution. The decline of trade policy uncertainty leads to the vigorous development of trade, the production efficiency of enterprises and technological innovation [49,50,51,52]. First, the inefficient use of non-clean energy leads to serious environmental pollution, the introduction of advanced technology and the emergence of new technology will increase enterprises’ production efficiency, the improvement in resource efficiency has an inhibitory effect on resource consumption [53], reducing the degree of pollution due to clean energy, to a certain extent. The emergence of technology to develop new and clean energy, this will solve the environmental problems caused by non-clean energy sources. Second, under the uncertainty of trade policy, the trade environment is stable, and higher environmental regulatory standards and environmental technologies will enter China [54]. At the same time, a large amount of foreign capital will enter the Chinese market, and sufficient capital will help enterprises increase R&D investment to improve energy conservation, emission reduction and pollution treatment technologies, so as to reduce environmental pollution. Finally, the reduction of trade policy uncertainty will bring a stable trade environment, and foreign investors are more willing to invest in China. Foreign direct investment, in addition to its influence on enterprises, also has a certain influence on the government. Local policy, in order to create a good investment environment, will increase spending on environmental protection, strengthening environmental protection and supervision. Thus, the environment can be improved [55]. 

## 7. Conclusions

This paper studies trade policy uncertainty and environmental pollution. First, due to data availability and other reasons, the existing research samples are basically at national and provincial levels, and the environmental pollution measurement index mostly deals with SO_2_, CO_2_ and NOX, etc. Few papers have adopted a more accurate index to measure PM_2.5_. Second, existing research on trade uncertainty pays more attention to enterprises’ behaviors such as import, export and innovation, and ignore the important impact of trade policy fluctuations on the environment. For this reason, the data on PM_2.5_ concentration in China from 1998 to 2007 are used in this paper. The internal relationship between trade policy uncertainty and environmental pollution in China is systematically discussed by using the difference-in-difference method. This is the first attempt to systematically study the impact and transmission mechanism of trade policy uncertainty on Chinese environmental quality, and the heterogeneity is analyzed using spatial characteristics and city size. The research findings of this paper are as follows: first, the reduction of trade policy uncertainty can significantly improve environmental quality, and this conclusion is still valid after a series of effectiveness and robustness tests. Second, trade policy uncertainty can effectively improve environmental quality through the industrial structure and technological effects. Third, in terms of heterogeneity analysis, first of all, from the perspective of spatial distribution characteristics, the reduction of trade policy uncertainty has a significant positive effect on environmental improvement in western China, while it has no significant effect on environmental improvement in central and eastern China. Secondly, from the perspective of city size, the reduction of trade policy uncertainty has a much greater effect on environmental improvement in small cities than in large and medium-sized cities.

The results of this study have important policy implications. First of all, faces with the severe current international situation, China has expanded its opening-up and firmly upheld an open world economy and multilateral trading system. This paper concludes that a stable trade environment can help improve environmental quality. In addition to the economic growth mechanism identified by classical literature, trade can also improve the welfare level of Chinese society by improving environmental conditions. Second, government should balance regional development, provide adequate financial support to underdeveloped areas and raise wide awareness of local development. Finally, the Chinese government should also advocate clean energy and advanced technology. At the same time, China must also lead local enterprises to establish the right type of development according to local conditions, gradually eliminate the traditional industries that have high emissions, high pollution and high energy consumption, and promote the industrial structure towards a green upgrade.

This article only explores the problem of pollution at the prefecture level and city level. China’s cities have had wide coverage, but reports of the county situation are more complicated. Studies at the county level may have more unexpected results; in terms of the future, scholars should continue to enrich the research in this field.

## Figures and Tables

**Figure 1 ijerph-19-02150-f001:**
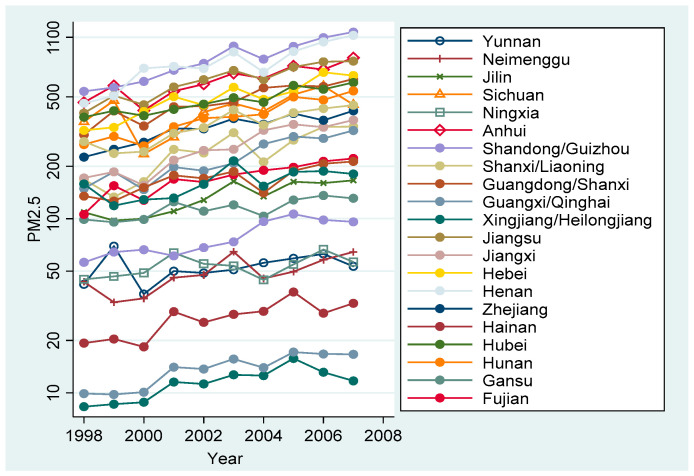
Variation trend of PM_2.5_ concentration in each province. Note: Data come from the China City Statistical Yearbook.

**Figure 2 ijerph-19-02150-f002:**
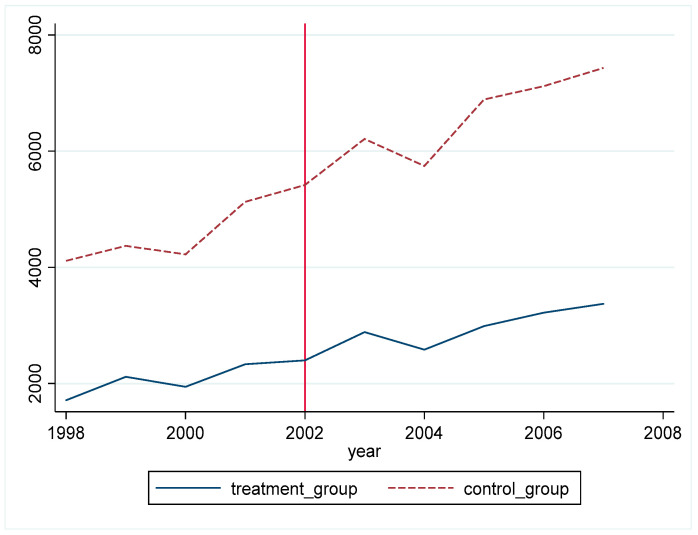
Change trend of PM_2.5_ concentration after grouping.

**Figure 3 ijerph-19-02150-f003:**
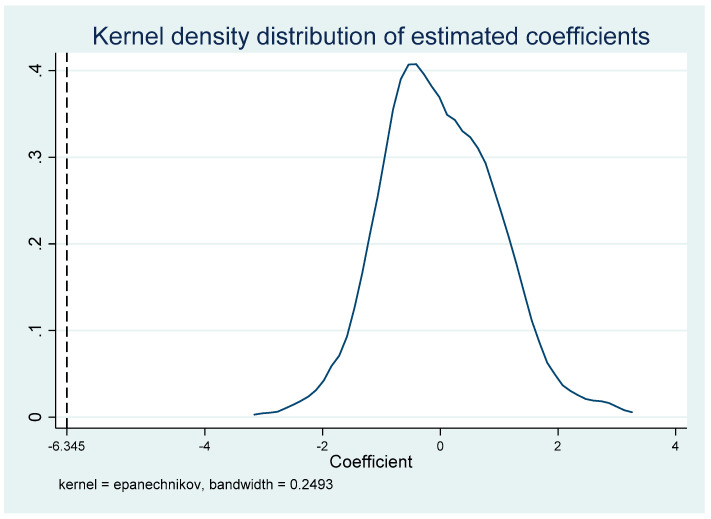
Placebo test.

**Table 1 ijerph-19-02150-t001:** Data and statistical description of variables.

Variable	Source of Raw Data	Observation	Average	SD	Min	Max
PM_2.5_	Ma et al. [38]	2460	38.3348	28.7365	1.2111	519.887
TPU	Chinese Industrial Enterprise database	2460	0.3262	0.2161	0	1.7808
K/L	China City Statistical Yearbook	2363	10.3893	0.9890	6.0551	12.8785
Pop	China City Statistical Yearbook	2364	5.8224	0.7811	2.3026	9.3557
Manu	China City Statistical Yearbook	2379	46.0783	10.9552	18.5	89.7
Fdi	China City Statistical Yearbook	2290	8.6271	1.9206	2.3026	13.4821
p. c. GDP	China City Statistical Yearbook	2386	5.8155	0.9392	1.9491	8.8691
Land	China City Statistical Yearbook	2365	13.9334	16.6423	0.236	253.356
Output	China City Statistical Yearbook	2150	14.8557	1.2630	11.0396	18.8850

Note: The data were obtained by manual sorting and calculation by the author. Due to statistical problems in some years in the urban yearbook, some data are missing. Among them, K/L, Pop, FDI, p. c. GDP and Output all adopt logarithmic form.

**Table 2 ijerph-19-02150-t002:** Baseline regression results.

PM_2.5_								
	(1)	(2)	(3)	(4)	(5)	(6)	(7)	(8)
${\mathrm{TPU}}_{jt}*Post02_{t}$	−5.295 **	−5.569 **	−5.599 **	−5.757 **	−5.582 **	−5.737 **	−5.658 **	−6.345 **
	(−2.20)	(−2.19)	(−2.20)	(−2.26)	(−2.26)	(−2.34)	(−2.29)	(−2.56)
Fdi		0.105	0.106	0.0927	−0.0344	0.0811	0.0612	0.143
		(0.31)	(0.32)	(0.28)	(−0.10)	(0.26)	(0.19)	(0.44)
Pop			0.686	0.503	0.455	2.056	4.608 *	5.787 **
			(0.40)	(0.28)	(0.26)	(1.07)	(1.77)	(2.16)
Manu				0.126 *	0.0817	0.202 ***	0.192 **	0.244 ***
				(1.83)	(1.23)	(2.70)	(2.60)	(2.80)
K/L					1.426 **	1.584 **	1.598 **	1.528 *
					(2.13)	(2.40)	(2.42)	(1.88)
p.c.GDP						−6.286 ***	−5.476 **	−5.572 **
						(−3.12)	(−2.56)	(−2.40)
Land							0.425	0.548 **
							(1.35)	(2.02)
Output								−1.624
								(−0.80)
_cons	39.37 ***	38.42 ***	34.41 ***	29.88 ***	18.44	0.957	−19.38	−4.988
	(83.67)	(13.21)	(3.35)	(2.79)	(1.52)	(0.07)	(−0.99)	(−0.14)
Year FE	YES	YES	YES	YES	YES	YES	YES	YES
Prefecture FE	YES	YES	YES	YES	YES	YES	YES	YES
N	2460	2289	2289	2280	2279	2279	2279	2080
R^2^	0.8774	0.8780	0.8780	0.8778	0.8781	0.8787	0.8789	0.8785

Note: ***, **, * represent the significance level of 1%, 5% and 10% respectively; the values in brackets are the corresponding T-statistics under robust standard error of clustering at the city level.

**Table 3 ijerph-19-02150-t003:** Validity test.

PM_2.5_		
	(1) Dynamic Effect	(2) Expectation Effect
year1999*TPU	−7.905	
	(−1.11)	
Year2000*TPU	−10.30	
	(−1.49)	
Year2001*TPU	−9.833	
	(−1.48)	
Year2002*TPU	−12.80 *	
	(−1.94)	
Year2003*TPU	−23.75 **	
	(−1.97)	
Year2004*TPU	−11.48 *	
	(−1.67)	
Year2005*TPU	−15.20 **	
	(−2.28)	
Year2006*TPU	−15.37 **	
	(−2.33)	
Year2007*TPU	−15.15 **	
	(−2.27)	
${\mathrm{TPU}}_{jt}*Post02_{t}$		−6.622 ***
		(−2.67)
${\mathrm{TPU}}_{jt}*PreReform$		−0.789
		(−0.26)
_cons	−1.430	−4.818
	(−0.04)	(−0.14)
Control variables	YES	YES
Year FE	YES	YES
Prefecture FE	YES	YES
N	2080	2080
R^2^	0.8786	0.8784

Note: ***, ** and * represent significant levels of 1%, 5% and 10%, respectively; the values in brackets are the corresponding T-statistics under robust standard error of clustering at the city level.

**Table 4 ijerph-19-02150-t004:** Robustness test.

	(1)	(2)	(3)	(4)	(5)	(6)	(7)	(8)	(9)	(10)	(11)
	TPU1	TPU2	TPU3	Province FE	SE Clustered at Province Level	Winsorize 5%	Truncate 5%	Expand the Sample to 2013	Trade World Weighted	Trade World Unweighted	Add Energy Variable
${\mathrm{TPU}}_{jt}1*Post02_{t}$	−7.399 **										
	(−2.11)										
${\mathrm{TPU}}_{jt}2*Post02_{t}$		−5.899 **									
		(−2.02)									
${\mathrm{TPU}}_{jt}3*Post02_{t}$			−5.185 *								
			(−1.95)								
${\mathrm{TPU}}_{jt}*Post02_{t}$				−6.345 **	−6.345 **	−4.259 **	−3.775 **	−6.167 **	−8.367 **	−5.005 **	−7.098 **
				(−2.55)	(−2.24)	(−2.34)	(−2.23)	(−2.29)	(−2. 59)	(−2. 19)	(−2.79)
_cons	−4.984	−5.080	−5.139	−4.988	−4.988	−17.31	−16.35	30.56 **	−10.61	−38. 48	5.101
	(−0.14)	(−0.15)	(−0.15)	(−0.14)	(−0.10)	(−0.61)	(−0.59)	(2.51)	(−0.30)	(−0. 92)	(0.14)
Control variables	YES	YES	YES	YES	YES	YES	YES	YES	YES	YES	YES
Energy consumption	NO	NO	NO	NO	NO	NO	NO	NO	NO	NO	YES
Year FE	YES	YES	YES	YES	YES	YES	YES	YES	YES	YES	YES
Prefecture FE	YES	YES	YES	YES	YES	YES	YES	YES	YES	YES	YES
Province FE	NO	NO	NO	YES	NO	NO	NO	NO	NO	NO	NO
N	2080	2080	2080	2080	2080	2080	1865	3519	2080	1696	
R^2^	0. 8784	0.8784	0.8767	0.8785	0.9345	0.9185	0.9135	0.9320	0.8785	0.8694	

Note: ** and * represent significant levels of 5% and 10% respectively; except column (5), the values in brackets are t statistics corresponding to a robust standard error of clustering at city level.

**Table 5 ijerph-19-02150-t005:** Bartik IV.

	(1) First Stage Regression	(2) Second Stage Regression
	TPU*_jt_***Post*02*_t_*	PM_2.5_
Bartik IV	1.048 ***	−86.847 *
	(2.85)	(−1.70)
_cons	0.135	−12.646 ***
	(0.32)	(5.22)
Control variables	YES	YES
Year FE	YES	YES
Prefecture FE	YES	YES
N	2080	2080
R^2^	0.8066	0.2133

Note: *** and * represent significant levels of 1% and 10% respectively.

**Table 6 ijerph-19-02150-t006:** Heterogeneity test results.

	(1)	(2)	(3)	(4)	(5)
	Eastern City	Central City	Western City	Large and Medium-Sized City	Small City
${\mathrm{TPU}}_{jt}*Post02_{t}$	−6.488	−5.675	−7.822 *	1.593	−7.359 ***
	(−1.59)	(−1.34)	(−1.84)	(0.25)	(−2.74)
_cons	−49.02	−197.1	−41.32	124.1	−24.34
	(−1.23)	(−1.35)	(−0.45)	(1.29)	(−0.63)
Control variables	YES	YES	YES	YES	YES
Year FE	YES	YES	YES	YES	YES
Prefecture FE	YES	YES	YES	YES	YES
N	847	793	440	404	1676
R^2^	0.9313	0.9403	0.7614	0.9501	0.8566

Note: *** and * represent significant levels of 1% and 10%, respectively; the values in brackets are the corresponding T-statistics under robust standard error of clustering at the city level.

**Table 7 ijerph-19-02150-t007:** Test results of influencing mechanism.

	(1)	(2)
	Technical Effect	Industrial Structure Effect
${\mathrm{TPU}}_{jt}*Post02_{t}$	0.580 *	1.273 **
	(1.68)	(2.07)
_cons	−2.328	62.09 ***
	(−0.83)	(5.22)
Control variables	YES	YES
Year FE	YES	YES
Prefecture FE	YES	YES
N	2080	2080
R^2^	0.9754	0.9393

Note: ***, **, * represent the significance level of 1%, 5%, 10%, respectively; The values in brackets are the corresponding T-statistics under robust standard error of clustering at the city level.

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
