# Peer review of "Does Trade Policy Uncertainty Exacerbate Environmental Pollution?—Evidence from Chinese Cities"

_ijerph, 2022, doi:10.3390/ijerph19042150_

Round 1

Reviewer 1 Report

I like the paper. It examines the association between pollution and the elimination of trade uncertainty that heterogeneously impacted industries in China over its WTO accession. The main problem of the paper is that there is no clear eyeball evidence of DID from Figure 2. Please check the following to justify your findings

  1. There are good progress in DID method in econometrics. Take a look at DeChaisemartin and Dhaultifeduille (AER 2020). That's a good starting point to introduce the updated DID models.
  2. Bartik (Shift-Share instrument) index now has a good foundation for its validity. Please check Adao et al (QJE 2019), Borusyak et al (REStud 2021), and Goldsmith-Pinkham et al (AER 2020).
  3. I think the authors should include some energy source variables (share of coal in power generation at the city level). Without the dirtiest source of air pollution, coal, it is not convincing. 
  4. Please check that low trade uncertainty industry located inland and high trade uncertainty industry located coastal area. Give more intuitive discussions at the regional level.

Author Response

  We thank you very much for the comments and suggestions. The comments and suggestions are valuable and very helpful for revising and improving our manuscript. We have made revisions according to the referees’comments and suggestions, as described in the authors’ response. According to your suggestion, the modifications are as follows:

1.About Figure2

Figure2 is used to judge the relationship between trade policy uncertainty and pollution preliminary, on the other hand it can be roughly judged that the two groups of cities basically meet the parallel trend hypothesis, this is only a simple judgment through image. We further made a dynamic effects to test parallel trend assumptions, all the above contents are supplemented in detail in the description of Figure2.

2.About energy source variables

As for energy variables, we tried to solve the problem by using coal consumption for robustness test, and the results were still robust. See 4.3 for specific results

3.About the standardization of DID method

After carefully reading the references given by experts, this paper draws on the practice of Bartik(1991) to construct a "Bartik Instrument".(TPU of first order lag times TPU of first order difference in time),Then we perform 2SLS two-stage instrumental variable estimation. Unfortunately, after multiple adjustments, the regression coefficient of Bartik's instrumental variables on PM2.5 is still not significant, so this part of the results are not reflected in the article. The results are shown below:

(1)

PM2.5

Bartik IV

-46.194

(-0.65)

CV

YES

N

2081

R2

0.5174

Reviewer 2 Report

I have read the paper “Does Trade Policy Uncertainty Exacerbates Environmental Pollution?——Evidence from Chinese Cities”. The authors investigate the relationship between trade policy uncertainty and environmental pollutions, by using 246 prefecture-level cities panel data. The topic has practical significance. The structure arrangement is reasonable and the argumentation process is reliable.

I here provide several comments for the improvement of the paper. The authors need to update their literature to 2021, and 2022. The latest reference the authors cited was in 2020.

It is necessary for authors to read some of the latest literature to ensure that they have mastered the latest and cutting-edge knowledge points. Several newly published papers can be reviewed: The heterogeneous impact of environmental regulation on urban green scale economy: an empirical analysis based on city-level panel data in China; The effect of resource abundance on Chinese urban green economic growth: A regional heterogeneity perspective; etc.

Li, X., Hu, Z., & Zhang, Q. (2021). Environmental regulation, economic policy uncertainty, and green technology innovation. Clean Technologies and Environmental Policy, 23(10), 2975-2988.

Li, X. L., Li, J., Wang, J., & Si, D. K. (2021). Trade policy uncertainty, political connection and government subsidy: Evidence from Chinese energy firms. Energy Economics, 99, 105272.

Shabir, M., Ali, M., Hashmi, S. H., & Bakhsh, S. (2021). Heterogeneous effects of economic policy uncertainty and foreign direct investment on environmental quality: Cross-country evidence. Environmental Science and Pollution Research, 1-16.

Bhowmik, R., Syed, Q. R., Apergis, N., Alola, A. A., & Gai, Z. (2021). Applying a dynamic ARDL approach to the Environmental Phillips Curve (EPC) hypothesis amid monetary, fiscal, and trade policy uncertainty in the USA. Environmental Science and Pollution Research, 1-15.

Yuan, B., Li, C., & Xiong, X. (2021). Innovation and environmental total factor productivity in China: the moderating roles of economic policy uncertainty and marketization process. Environmental Science and Pollution Research, 28(8), 9558-9581.

Anser, M. K., Apergis, N., & Syed, Q. R. (2021). Impact of economic policy uncertainty on CO 2 emissions: evidence from top ten carbon emitter countries. Environmental Science and Pollution Research, 1-10.

Abbasi, K. R., & Adedoyin, F. F. (2021). Do energy use and economic policy uncertainty affect CO 2 emissions in China? Empirical evidence from the dynamic ARDL simulation approach. Environmental Science and Pollution Research, 28(18), 23323-23335.

Song, Y., Hao, F., Hao, X., & Gozgor, G. (2021). Economic policy uncertainty, outward foreign direct investments, and green total factor productivity: evidence from firm-level data in China. Sustainability, 13(4), 2339.

Yu, J., Shi, X., Guo, D., & Yang, L. (2021). Economic policy uncertainty (EPU) and firm carbon emissions: Evidence using a China provincial EPU index. Energy Economics, 94, 105071.

That’s all. Good luck.

Author Response

Thanks for the your suggestions. After carefully reading your suggestions and reference materials, we cited the literature you provided, and we looked for more literatures and selected appropriate literatures to cite.The specific modifications can be see the article.

Reviewer 3 Report

The article deals with the extremely important and topical issues of the relationship between the uncertainty of trade policy and environmental pollution. The proposed research method allowed to achieve interesting results. Importantly, the text is not only scientific, but also utilitarian.

Some details, however, require some fine-tuning:

  1. It is worth defining the goal. From the content of the abstract, introduction and conclusions, I can guess what the purpose of the research was, however, the goal should be indicated - in the abstract and in the introduction.
  2. Complete the sources in introduction, especially with regard to numerical data, replace the footnotes with parentheses. The introduction contains a lot of data and statements and hardly any sources.
  3. Identify the research gap filled by the research. Perhaps it will help to clarify the goal of the research.
  4. In subsection 3.1, it is also worth mentioning the sources of numerical data.
  5. Indicate the limitation of research. Complex research is always associated with numerous limitations. Limitations may become a determinant for other researchers.
  6. Indicate possible directions for further research in this area.

Good luck!!!

Author Response

  We thank you very much for the comments and suggestions. The comments and suggestions are valuable and very helpful for revising and improving our manuscript. The specific modifications are as follows:

  1. Research objectives

In the abstract and introduction, we uses three questions to reflect the research objectives of this paper.

  1. Data sources

This paper further marks the sources of research data in a prominent position, so that readers can view them more clearly and quickly.

  1. The marginal contribution of this paper is explained in detail in the last part of the second Literature Review.
  2. The research outlook and future research direction have been supplemented at the end of this paper.

Round 2

Reviewer 1 Report

The paper should reflect recent development in DID or SSIV (Bartik) and test the hypothesis with new standard errors. 

Author Response

Thank you for your valuable advice! After careful and careful reading of your comments, we make the following changes, which are hereby stated:

  1. Standardization of DID method

Thank you very much for your advice. After repeated reading and understanding, the author finally drew on the practice of Bartik(1991) to construct a "Bartik Instrument" (product of first-order lag TPU and TPU's first-order difference in time), and then carried out 2SLS two-stage tool variable estimation. After continuous adjustment and correction of errors, satisfactory results were finally obtained. See 4.4 for specific results.

  1. Consider more comprehensive robustness tests

Inspired by your advice, we consulted numerous materials and searched a large number of databases, and finally found the statistics of coal consumption in China's Energy Statistical Yearbook. The author tried to conduct robustness test with coal consumption, and achieved good results. See 4.3 for the specific results.

  1. References

We look for more related new literature, and choose appropriate literature for reference, to ensure the effectiveness of the article.

  1. Research design

We modified the abstract and introduction to further highlight the research objectives, make the research ideas, design and conclusion more clear, and also show the marginal contribution of this paper more clearly. In addition, the author supplemented the research prospect.

Reviewer 2 Report

Accept.

Author Response

Dear expert:

     Hello!Thank you very much for your advice, we are deeply inspired from it, research in the future will learn from experience, further improve our research level, wish you happy life!